# MRSA and Skin Infections in Psoriatic Patients: Therapeutic Options and New Perspectives

**DOI:** 10.3390/antibiotics11111504

**Published:** 2022-10-28

**Authors:** Giulio Rizzetto, Elisa Molinelli, Giulia Radi, Oscar Cirioni, Lucia Brescini, Andrea Giacometti, Annamaria Offidani, Oriana Simonetti

**Affiliations:** 1Clinic of Dermatology, Department of Clinical and Molecular Sciences, Polytechnic University of Marche, 60126 Ancona, Italy; 2Clinic of Infectious Diseases, Department of Biomedical Sciences and Public Health, Polytechnic University of Marche, 60126 Ancona, Italy

**Keywords:** psoriasis, skin and soft tissue infections, wound healing, staphylococcal skin infection, MRSA, quorum sensing

## Abstract

Psoriatic patients present various infectious risk factors, but there are few studies in the literature evaluating the actual impact of psoriasis in severe staphylococcal skin infections. Our narrative review of the literature suggests that psoriatic patients are at increased risk of both colonization and severe infection, during hospitalization, by *S. aureus*. The latter also appears to play a role in the pathogenesis of psoriasis through the production of exotoxins. Hospitalized psoriatic patients are also at increased risk of MRSA skin infections. For this reason, new molecules are needed that could both overcome bacterial resistance and inhibit exotoxin production. In our opinion, in the near future, topical quorum sensing inhibitors in combination with current anti-MRSA therapies will be able to overcome the increasing resistance and block exotoxin production. Supplementation with Vitamin E (VE) or derivatives could also enhance the effect of anti-MRSA antibiotics, considering that psoriatic patients with metabolic comorbidities show a low intake of VE and low serum levels, making VE supplementation an interesting new perspective.

## 1. Introduction

Psoriasis is a chronic immune-mediated inflammatory disease that affects 1 to 2% of the population in Western countries [1,2]. On the one hand, mild and moderate forms can be treated with topical therapy, although corticophobia is increasingly common, especially among children [3]. On the other hand, severe forms require systemic therapy, which may include conventional immunosuppressive or immunomodulatory agents [4]. Psoriatic patients are potentially at increased risk of developing severe infections, including skin and skin structure infections (SSSI). This may be due to alterations in the immune response, immunosuppressive therapy and associated metabolic comorbidities, such as diabetes, overweight and obesity [5,6,7,8,9].

Patients with active disease present a localized impairment of the skin barrier, with reduced expression of tight junctions and altered production of ceramides. This may also lead to an impairment of the normal skin microbiota. Furthermore, a dysregulation of both adaptive and innate immunity is preponderant, resulting in a skin more susceptible to infectious agents [6,7]. Among these, *Staphylococcus aureus* seems to play a prominent role, especially considering the cases of severe infections requiring hospitalization [10]

Staphylococcal infections are among the most common complications of skin lesions [11,12] and contribute to slow wound healing [13]. In addition, the increasing antibiotic resistance of *S. aureus* has a great impact on public health [13], requiring new molecules that can overcome the resistance mechanisms of methicillin-resistant *Staphylococcus aureus* (MRSA) and promote skin healing [14,15,16,17].

In the literature, data on the risk of cutaneous staphylococcal infections in psoriatic patients are few and sometimes contradictory. We, therefore, performed a narrative review, aiming to provide an overview of the role of both infection and skin staphylococcal colonization in psoriatic patients. In addition, we assessed the available therapeutic options and provided our opinion on new therapeutic perspectives in cases of cutaneous MRSA infection for psoriatic patients.

### 1.1. S. aureus Skin Colonization in Psoriatic Patients

Psoriasis is an inflammatory disease with a complex pathogenesis, in which dendritic cells and macrophages, acting as antigen presenting cells (APCs), activate T helper (Th) lymphocytes. These tend to polarize their response into the Th1 type, producing interferon-γ (IFN-γ), and the Th17 type, producing IL-17, determining a complex immune interaction with macrophages, neutrophils and mast cells. This results in the release of inflammatory cytokines (such as TNF-α, IFN- γ, IL-12, IL-22, IL-17 and IL-23) leading to keratinocyte proliferation and loss of normal epidermal structure and functions [18]. 

Like other inflammatory skin diseases, there are emerging studies that correlate disease activity of psoriasis with staphylococcal superantigens (as toxic shock syndrome toxin, staphylococcal enterotoxin a/b and exfoliative toxin b), which may be a trigger in activating APCs and start the typical inflammatory response [18,19,20]. Skin colonization by *S. aureus* in psoriatic patients, with a consequent alteration of the local microbiota, appears to support the inflammatory pathogenesis of psoriasis by contributing to the alteration in the immune response [21,22]. Specifically, it was observed that in the skin of psoriatic patients, compared to healthy controls, there was, on the one hand, a reduction in *Cutibacterium acnes*, *Cutibacterium granulosum* and *Staphylococcus epidermidis* and, on the other hand, an increase in *Staphylococcus aureus* and *Staphylococcus pettenkoferi* [22].

Antimicrobial peptides (AMPs), such as β-defensin, S100 and cathelicidin LL-37, produced locally by keratinocytes and immune cells, also play a role in the pathogenesis of psoriasis, since they can modulate the inflammatory response and modulate cell proliferation. A role of LL-37 has been shown in promoting Th17 polarization by acting as a damage-associated molecular pattern (DAMP) enhancer, binding to self-DNA and self-RNA released by neutrophils in psoriatic plaques and activating toll-like receptors (TLRs) 7 and 8 of dendritic cells. Furthermore, LL-37 expression is increased in skin areas affected by psoriasis [23]. The normal skin microbiota, of which *S. epidermidis* is the dominant bacterium, tends to produce other AMPs (such as thiolactone-containing peptide) that control the growth of other bacteria, including *S. aureus* [24]. To the best of our knowledge, the relationship between *S. aureus* and LL-37 production in psoriatic plaques has not yet been clearly defined and represents an interesting new perspective for further studies.

However, the colonization of psoriatic patients by *S. aureus* was evaluated in a systematic review [25], showing a statistically significant 4.5-fold higher risk of being colonized on lesional skin compared to healthy subjects. In addition, MRSA appeared to colonize both the skin and nasal mucosa of psoriatic patients with a higher prevalence (pooled prevalence 8.6%) than healthy controls (2.6%), although this difference was not statistically significant due to the low sample size.

Although it appears that the pathogenic role of *S. aureus* is marginal in psoriasis, since more than one pathogen is involved at the same time in the lesional skin [26], some studies report that the interaction between *S. aureus* and the skin, involving both innate and adaptive immunity, may drive the imbalance in the T helper (Th) 1/Th17 response axis in psoriasis [27,28,29]. This hypothesis is supported in the study by *Balci* et al. [18], showing that there is an increased prevalence of toxigenic strains in the skin lesions of psoriatic patients compared to healthy subjects. *Atefi* et al. [30] also confirmed, in a case-control study, that psoriatic patients have a higher frequency of staphylococcal superantigens measured in serum. In particular, there was a statistically significant difference between psoriatic patients and controls, considering Toxic Shock Syndrome Toxin (TSST), 47% vs. 6%, respectively, and Enterotoxins (A, B, D), 48.8% vs. 6%, respectively.

These data may also have a practical implication, as the serum test for TSST has been shown to correlate with the form of chronic plaque psoriasis, and the authors suggest that adding adequate antibiotic therapy against *S. aureus* may be useful in addition to psoriasis therapy in the case of a positive serum test for the presence of staphylococcal superantigens. Further studies on the effect of *S. aureus* control in psoriatic patients are needed to confirm these hypotheses, considering that the risk of developing resistant strains is very high. New therapeutic approaches are, therefore, also needed to overcome possible antibiotic resistance.

### 1.2. S. aureus and SSSI in Psoriatic Patients

Psoriatic patients have multiple risk factors that may predispose them to severe infections, such as impaired immune response and systemic immunosuppressive therapies [31]. In an 11-year American national cross-sectional study including a sample of 2738 patients hospitalized primarily for psoriasis and 184,508 patients hospitalized with a secondary diagnosis of psoriasis, Hsu et al. [10] associated psoriasis with a statistically significant increased risk of multiple severe infections compared to hospitalized patients without psoriasis (OR 1.3, *p* < 0.0001). These included SSSIs, such as cellulitis and infections sustained by both MRSA and methicillin-sensitive *Staphylococcus aureus* (MSSA).

Psoriatic patients are, therefore, at increased risk of developing severe staphylococcal infections in the hospital setting. These also lead to a significant increase in costs ($13,291 ± $166 vs $11,003 ± $96) and in the length of hospitalization (6.6 ± 0.1 days vs 4.6 ± 0.03 days). A limitation of this study is the impossibility of correlating psoriasis severity with infectious risk and on-going therapy. However, another study in the literature reports that the severity of the Psoriasis Area and Severity Index (PASI) correlates significantly with the toxin-producing capacity of lesional strains of *S. aureus* [32].

Finally, other studies in the literature do not report an increase in serious staphylococcal infectious episodes in patients with moderate to severe psoriasis on therapy with ustekinumab [33] or adalimumab [34], suggesting that appropriate immunomodulatory therapy for psoriasis may reduce the risk of severe staphylococcal infections. These drugs, interleukin (IL) 12/23 and tumor necrosis factor (TNF) α inhibitors, respectively, do not, in fact, act as immunosuppressants but, rather, selectively modulate the immune response involved in the pathogenesis of psoriasis. The resulting sustained reconstitution of the integrity of the skin barrier may reduce the risk of developing *S. aureus* infections. However, these biological drugs may increase the risk of reactivation of latent chronic infections, as in the case of tuberculosis, HIV, HCV and HBV. There are no other specific data in the literature on serious *S. aureus* infections in psoriatic patients undergoing therapy with biologics, however, their use is contraindicated in the case of serious infections [35].

## 2. Management of *S. aureus* SSSI

In view of data on colonization and severe MSSA/MRSA infections in psoriatic patients, we believe that specific antimicrobial therapy should be considered in appropriate cases.

According to the Infectious Diseases Society of America (IDSA) guidelines, SSSIs should be divided into purulent, including cellulitis and erysipelas, and non-purulent, including abscesses [36]. For severe non-purulent SSSIs, vancomycin plus piperacillin/tazobactam is recommended as the first line, whereas cefazolin, ceftriaxone, penicillin and clindamycin are recommended for moderate forms. For purulent SSSI, vancomycin, tigecycline, linezolid, daptomycin, telavancin and ceftaroline can be used in severe forms, whereas doxycycline and trimethoprim-sulfamethoxazole are recommended for moderate forms [36].

If MSSA is isolated after skin swab, the antibiotics of choice are cefazolin and clindamycin for severe infections and cephalexin and dicloxacillin for moderate infections, with a duration of 7–10 days. In the event of MRSA isolation or high-risk infections, such as failure of first-line therapy, systemic inflammatory response syndrome (SIRS) or immunosuppressive conditions, as those associated with the treatment of psoriasis, anti-MRSA antibiotics should be added. These include vancomycin, the first-line therapy of serious hospital acquired MRSA infections, daptomycin, linezolid, tigecycline, ceftaroline, tedizolid, dalbavancin, oritavancin and telavancin [36,37].

In our opinion, in psoriatic patients with SSSIs requiring hospitalization, empirical therapy should consider the use of an anti-MRSA drug. Our review of the literature showed that cellulitis is the most frequently occurring skin infection in hospitalized psoriatic patients. As it is a non-purulent condition, guidelines recommend the use of vancomycin plus piperacillin/tazobactam as a first line. However, serum vancomycin levels and renal function must be monitored [38], making its use difficult in some patients. Among empirical second-line therapies, clindamycin may be ineffective in cases where the local resistance rate is greater than 10% [36]. Linezolid is an effective alternative, but it must be considered that neurotoxicity and myelosuppression may occur with prolonged treatment for more than 28 days [39]. 

Daptomycin is an option to consider, as it is not inferior to clindamycin and vancomycin in the treatment of SSSIs in pediatric trials [40,41]. Furthermore, in SSSIs requiring an improvement in wound healing, daptomycin has been shown to be effective both in eradicating the infection and in promoting skin healing through immunomodulatory effects, firstly by reducing local levels of interleukin (IL)-6 and matrix metalloproteinase (MMP)-9 and, secondly, by increasing metallopeptidase inhibitor (TIMP)-1, epidermal growth factor receptor (EGFR) and fibroblast growth factor (FGF)-2 [42,43]. However, a careful evaluation of risks and benefits must be made prior to the administration of daptomycin, also considering the possibility of side effects such as increased creatine phosphokinase and acute eosinophilic pneumonia [44].

Tigecycline is also an antibiotic that has been shown to be particularly effective in a mouse model for the control of many MRSA strains collected from skin wounds of hospitalized patients, also demonstrating an interesting synergistic effect with daptomycin and rifampicin, respectively, superior to single treatments [45]. This could represent a therapeutic option in cases where pathogens are resistant to single treatment.

Among the lipoglycopeptides, dalbavancin plays an important role considering its efficacy in treating SSSIs and its long half-life, which allows weekly dosing [46]. Again, in a comparison on MRSA-infected murine wounds between vancomycin and daptomycin, the latter demonstrated a greater reduction in bacterial load and better wound healing, with higher local EGFR and VEGF levels than vancomycin [47]. In a recent meta-analysis of 7289 patients [48], it was reported that telavancin, dalbavancin and oritavancin were not inferior to vancomycin in the treatment of SSSIs, and, in particular, telavancin showed greater efficacy in the eradication of MRSA, even if with not statistical significance (*p* = 0.06). On the other hand, telavancin showed a higher number of adverse effects than the other treatments (OR:1.24 *p* < 0.01).

Despite these therapeutic options, the development of new molecules, also to be combined with already approved drugs, is necessary to combat increasing resistance and improve skin healing [49].

## 3. New Perspectives on the Treatment of MRSA-Induced SSSI

More and more cases of MRSAs resistant even to specific therapy are reported in the literature, highlighting the urgency of evaluating new antimicrobial agents that may increase treatment options in the future [50,51,52,53]. In our opinion, new perspectives consist of the application of these new molecules in combination with the currently available drugs against MRSA both for the treatment of severe infections and inhibition of staphylococcal quorum sensing (QS) [49], attempting to block the production of exotoxins and potentially target the infectious trigger of psoriasis (Table 1).

Inhibitors of staphylococcal QS include RNA III inhibiting peptide (RIP), a seven amino acids peptide that blocks RNAIII production, reducing the expression of virulence factors [54,55]. RIP-soaked foam was used topically in combination with systemic teicoplanin in a mouse model with MRSA-infected wounds demonstrating a significant reduction in bacterial load in the combination group compared to single treatments [56]. This is an example of how a topical treatment, with a localized effect, can effectively be associated with conventional MRSA therapy. Topical RIP has also been used on humans in two cases of an MRSA-infected diabetic foot ulcer in combination with daptomycin and linezolid [57]. Other topical QS inhibitors used against MRSA, derived from RIP, are FS10 associated with tigecycline [58], F1 and F12 [59]. However, these anti-microbial peptides (AMPs) have not yet been tested in humans, so RIP is currently the molecule with the highest potential in the near future.

Other topical AMPs that have been reported to be effective on the QS of MRSA in mouse models. In monotherapy, as well, they are autoinducing peptide (AIP)-I, an inhibitor of MRSA *Agr* signalling [60], 430D-F5, a flavone-rich extract inhibitor of all *S. aureus Agr* alleles [61], apicidin [62] and temporin A [63]. However, these molecules need to be tested in humans to assess their actual clinical use.

The combination of anti-MRSA antibiotics and other systemic molecules may also increase therapeutic efficacy, such as tocotrienols supplementation in daptomycin therapy. This showed, in a mouse model with MRSA-infected lesions, a greater bactericidal capacity than single treatments [64]. In another study, the bactericidal enhancer action of vitamin E (VE) in combination with tigecycline and daptomycin in the treatment of murine cutaneous MRSA infections is reported [65]. In addition, tigecycline showed increased skin wound healing through modulation of metalloproteinases-9 [66]. In our opinion, supplementing VE or its derivatives could be applicable in the near future for cases of increasing MRSA resistance in the SSSIs of psoriatic patients, considering the ready accessibility of this kind of supplementation. Furthermore, in psoriatic patients with metabolic comorbidities and, thus, higher infective risk, it is reported that there is a low intake of VE [67] and low serum levels [68], making VE supplementation an interesting new perspective.

A new low-cost therapeutic perspective is also represented by phototherapy with a combination of two wavelengths, 460 nm and 405 nm, which showed itself to be effective in controlling MRSA infection in a mouse model [69]. Phototherapy also acts by restoring the barrier function of psoriatic lesional skin, with an improvement in both transepidermal water loss (TEWL), stratum corneum hydration (SCH), pH and elasticity [70]. This could contribute, in our opinion, to the reduction in the risk of colonization and infection by *S. aureus*, restoring physical skin barrier function and also indirectly acting on local microbiota alterations in the psoriatic plaque [71].

Finally, the use of nanotechnology as a carrier for various drugs [72] could also help in controlling MRSA infections in psoriatic patients. Among the molecules that, in our opinion, could be more easily used in the near future, mainly due to their low cost, are zinc oxide nanoparticles (ZnO-NPs), which act in vitro by producing reactive oxygen species and inhibiting amino acid synthesis in *S. aureus* strains [73] and nickel nanoparticles (NiNPs), which act by altering the structure of bacterial proteins and bacterial cell membranes and inhibiting DNA replication in MRSA [74]. However, the lack of human studies and the possible risk of increased penetration into the skin, with further impairment of the skin barrier, require further studies before use in psoriatic patients.

**Table 1 antibiotics-11-01504-t001:** Summary of new perspectives for the treatment of MRSA in psoriatic patients.

Treatment	Experimental Model	Advantage/Disadvantage
Quorum sensing inhibitors		
RIP [56]	Murine model, MRSA infected skin wound Topical RIP (20 mcg), teicoplanin i.p., allevyn, allevyn + teicoplanin i.p., topical RIP + teicoplanin i.p.	Topical RIP enhances teicoplanin effect against MRSA Better wound healing with topical RIP + teicoplanin (epithelial, granulation, collagen scores, microvessel density and VEGF expression)
RIP [57]	Case 1: 56 y.o. patient with diabetic foot ulcer, MRSA infection. Linezolid 600 mg 2/die + daily topical RIP (1 mg/cc) Case 2: 48 y.o. patient with leg gangrene/diabetic ulcer, MRSA infection. Daptomycin 6 mg/kg daily + topical RIP (1 mg/cc)	Rapid improvement, healing in 12 (Case 1) and 24 (Case 2) weeks, avoiding amputation. Excellent synergy with linezolid and daptomycin First case reports on human patients with severe infections Best candidate for future applications
FS10 [58]	Murine model, MRSA and MSSA-infected wounds Topical FS10 (20 mcg) + tigecycline i.p. (7 mg/kg) vs. monotherapy vs. untreated vs. uninfected	Better infection control and wound healing with combination FS10 + tigecycline Synergism with tigecycline No data on humans
Vit. E and derivates		
Tocotrienols (T3s) [64]	Murine model, MRSA infected wounds T3s, daptomycin i.p, T3s + daptomycin i.p.	Better bactericidal effect T3s + daptomycin Synergism with daptomycin Well-known and low-cost molecule No data on humans
Vitamin E (VE) [65]	Murine model, MRSA infected wounds VE, tigecycline i.p, VE + tigecycline i.p.	Bactericidal enhancer action VE + tigecycline Synergism with tigecycline Well-known and low-cost molecule No data on humans Low serum level [70] and intake [69] of VE in psoriatic patients with metabolic comorbidities New possible strategy for reducing MRSA infection risk
Phototherapy		
Combination of 2 wavelengths 460 nm and 405 nm [69]	Murine model, MRSA infected wounds 460 nm (360 J/cm^2^), 405 nm (342 J/cm^2^)	Bactericidal MRSA action No apoptosis of skin cells Low-cost therapy Possible skin barrier improvement in psoriatic patients [72] No data on humans Need to evaluate synergism with other MRSA antibiotics
Nanotechnology		
Zinc oxide nanoparticles (ZnO-NPs) [73]	In vitro MRSA strains	Low-cost molecules Need to evaluate synergism with other MRSA antibiotics Lack of human studies Possible risk of increased penetration and skin barrier impairment
Nickel nanoparticles (NiNPs) [74]	In vitro MRSA strains	Low-cost molecules Need to evaluate synergism with other MRSA antibiotics Lack of human studies Possible risk of increased penetration and skin barrier impairment

RNA III inhibiting peptide (RIP), intra peritoneal (i.p.). years old (y.o.).

## 4. Conclusions

Our narrative review of the literature shows that psoriatic patients may have a higher frequency of colonization by *S. aureus* than healthy controls. In addition, hospitalized psoriatic patients are at greater risk of developing a staphylococcal skin infection than individuals without psoriasis. For this reason, subjects with active infection must be treated appropriately, also considering the increased risk of MRSA infection. In the future, new molecules, including AMPs, and new strategies, also in combination with current anti-MRSA therapies, could be used to better control the colonization of multi-resistant strains and eradicate exotoxin-producing *S. aureus* strains, with a potential impact on the pathogenesis of psoriasis. 

To the best of our knowledge, there is a lack of crucial information in the literature regarding the correlation between psoriasis and *S. aureus*. Further studies are needed to better explore the role of staphylococcal infection and colonization in the pathogenesis of psoriasis, especially at the molecular level, potentially allowing new mechanisms to be discovered that can be used in the therapeutic approach.

## Data Availability

Not applicable.

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
