# Peer review of "MRSA and Skin Infections in Psoriatic Patients: Therapeutic Options and New Perspectives"

_antibiotics, 2022, doi:10.3390/antibiotics11111504_

Round 1

Reviewer 1 Report

The authors present a narrative review manuscript entitled “MRSA and skin infections in psoriatic patients: therapeutic options and new perspectives”.  The aim of this review was to provide an overview of the role of both infection and skin staphylococcal colonization in psoriatic patients. The authors also present a summary of the available therapeutic options and provide their opinion on new therapeutic perspectives in cases of cutaneous MRSA infection for psoriatic patients.

General concept comments

The authors use the following keywords for search in PubMed: psoriasis, staphylococcal infections, SSSI, MRSA, S. aureus, treatment, and quorum sensing. They selected 67 manuscripts to review but the cited bibliography is composed of several narrative reviews, please cite the original manuscripts.

The introduction to the description of psoriasis is general and incomplete, I consider that a better description is necessary to understand the risk factors for colonization and infection. Immune cells, cytokines, and antimicrobial peptides are implicated in the inflammatory disease, as well as, the skin barrier and microbiome alterations.

In several sections, the authors present results in practically the exact same way that were presented in the original papers.

Example 1:

Review:

“In a systematic review by Ng et al. [23] of 21 studies, it was found that psoriatic patients had a 4.5-fold higher risk of being colonized on lesional skin and in the nares by S. aureus compared to healthy subjects, relative risk (RR) 5.54, 95% confidence interval (CI) 3.21-9.57. Considering 8 comparative studies, the prevalence of staphylococcal skin colonization in psoriatic patients was 36.6% (95% CI 18.3-54.8), vs. 5.1% in healthy subjects, (95% CI 3.0-8.5). In addition, MRSA appeared to colonize both the skin and nasal mucosa of psoriatic patients with a higher prevalence (pooled prevalence 8.6%) than healthy controls (2.6%), although this difference was not statistically significant (p=0.74)”.

Original article, Abstract section:

“Overall we identified 21 eligible studies, of which 15 enrolled one or more comparison groups. The pooled prevalence of staphylococcal colonization in patients with psoriasis was 35·3% [95% confidence interval (CI) 25·0–45·6] on lesional skin and 39·2% (95% CI 33·7–44·8) in the nares. Patients with psoriasis were 4·5 times more likely to be colonized by S. aureus than healthy controls were on the skin (RR 5·54, 95% CI 3·21–9·57) and 60% more in the nares (RR 1·60, 95% CI 1·11–2·32). Cutaneous and nasal colonization by meticillin-resistant S. aureus also appeared higher in patients with psoriasis (pooled prevalence 8·6%) than in healthy controls (2·6%), yet the difference was not statistically significant (P = 0·74)”. 

Example 2:

Review:

“Atefi et al. [29] also confirmed in a case-control study that psoriatic patients have a higher frequency of staphylococcal superantigens measured in serum. In particular, Toxic Shock Syndrome Toxin (TSST) was detected in 47% (20/41 cases) vs. 6% (1/28 controls), (p = 0.000), as well as Enterotoxins (A, B, D) were detected in 48.8% (21/41 cases) vs. 6% (1/21 controls), (p = 0.000)”

Original article, Abstract section:

“TSST (toxic shock syndrome toxin) was detected in 47% (20/41) of cases and in 6% (1/28) of the controls with a significant difference. (P value = 0.000) Entrotoxins (A, B, D) were detected in the sera of 48.8% (21/41) of cases; and only 6 %(1/21) of controls, showed significant differences (P value = 0.000)

Results that are non-significative should not be addressed in a review unless controversy is reported in that regard. The authors should extract the main findings in each manuscript cited.  

I suggest the use of tables to summarize results, this will help to organize information and makes the manuscript text more fluent.

The authors don’t mention the alterations in the epidermis that is also a risk factor for infection. A recent study shows that psoriatic plaques present epidermal barrier dysfunction and the use of phototherapy may improve the epidermal barrier in psoriatic patients (Montero-Vilchez T, et al., 2021. Epidermal barrier changes in patients with psoriasis: The role of phototherapy. Photodermatol Photoimmunol Photomed. 2021 Jul;37(4):285-292). Phototherapy is also one of the therapeutics in development for the treatment of Staphylococcus aureus skin infections, so may be one of the new therapeutic perspectives to be considered in the review.

Another gap in new therapeutic perspectives in this manuscript is nanotechnology, there is extended literature on the use of different nanoparticles for the treatment of Staphylococcus aureus infections, and a recent study present this technology also for drug administration in psoriatic patients (Petit RG, et al.; 2021. Psoriasis: From Pathogenesis to Pharmacological and Nano-Technological-Based Therapeutics. Int J Mol Sci. 2021 May 7;22(9):4983).

I find the topic of the review extremely interesting but the manuscript was not rigorously organized, and lack critical information concerning the pathology of psoriasis that could help understand the link between Staphylococcus aureus infections and psoriasis, whether psoriasis favors the colonization and infection or not; whether the bacteria enhance psoriasis or not, the mechanism that are or could be involved, etc. All the questions that do not find answers in the literature could be presented as interesting fields to explore in conclusion and perspectives.  

Author Response

The authors present a narrative review manuscript entitled “MRSA and skin infections in psoriatic patients: therapeutic options and new perspectives”.  The aim of this review was to provide an overview of the role of both infection and skin staphylococcal colonization in psoriatic patients. The authors also present a summary of the available therapeutic options and provide their opinion on new therapeutic perspectives in cases of cutaneous MRSA infection for psoriatic patients.

General concept comments

The authors use the following keywords for search in PubMed: psoriasis, staphylococcal infections, SSSI, MRSA, S. aureus, treatment, and quorum sensing. They selected 67 manuscripts to review but the cited bibliography is composed of several narrative reviews, please cite the original manuscripts. 

RE: We tried to refer to the original manuscripts cited in the reviews, however we consider it useful that the reviews are also reported as, although narrative, they may express opinions that we consider interesting in relation to the topic

The introduction to the description of psoriasis is general and incomplete, I consider that a better description is necessary to understand the risk factors for colonization and infection. Immune cells, cytokines, and antimicrobial peptides are implicated in the inflammatory disease, as well as the skin barrier and microbiome alterations.

RE:Done.

“Psoriasis is an inflammatory disease with a complex pathogenesis, in which dendritic cells and macrophages, acting as antigen presenting cells (APCs), activate T helper (Th) lymphocytes. These tend to polarize their response into the Th1 type, producing interferon-γ (IFN-γ), and the Th17 type, producing IL-17, determining a complex immune interaction with macrophages, neutrophils and mast cells. This results in the release of inflammatory cytokines (such as TNF-α, IFN- γ, IL-12, IL-22, IL-17, and IL-23) leading to keratinocyte proliferation and loss of normal epidermal structure.[18]

Like other inflammatory skin diseases, there are emerging studies that correlate disease activity of psoriasis with staphylococcal superantigens (as toxic shock syndrome toxin, staphylococcal enterotoxin a/b, and exfoliative toxin b), which may be a trigger in activating APCs and start the typical inflammatory response. [18-20]. Skin colonization by S. aureus in psoriatic patients, with a consequent alteration of the local microbiota, appears to support the inflammatory pathogenesis of psoriasis by contributing to the alteration of the immune response [21,22]. Specifically, it was observed that in the skin of psoriatic patients, compared to healthy controls, there was on the one hand a reduction in Cutibacterium acnes, Cutibacterium granulosum, and Staphylococcus epidermidis, and on the other hand an increase in Staphylococcus aureus and Staphylococcus pettenkoferi. [22]

Antimicrobial peptides (AMPs), such as β-defensin, S100, and cathelicidin LL-37, produced locally by keratinocytes and immune cells, also play a role in the pathogenesis of psoriasis, since they can modulate the inflammatory response and modulate cell proliferation. A role of LL-37 has been shown in promoting Th17 polarization by acting as a damage-associated molecular pattern (DAMP) enhancer, binding to self-DNA and self-RNA released by neutrophils in psoriatic plaques and activating tool-like receptors (TLRs) 7 and 8 of dendritic cells. Furthermore, LL-37 expression is increased in skin areas affected by psoriasis. [23] The normal skin microbiota, of which S. epidermidis is the dominant bacterium, tends to produce other AMPs (thiolactone-containing peptide) that control the growth of other bacteria, including S. aureus. [24] At the best of our knowledge, the relationship between S. aureus and LL-37 production in psoriatic plaques has not yet been clearly defined and represents an interesting new perspective for further studies.”

In several sections, the authors present results in practically the exact same way that were presented in the original papers. 

Example 1:

Review: 

“In a systematic review by Ng et al. [23] of 21 studies, it was found that psoriatic patients had a 4.5-fold higher risk of being colonized on lesional skin and in the nares by S. aureus compared to healthy subjects, relative risk (RR) 5.54, 95% confidence interval (CI) 3.21-9.57. Considering 8 comparative studies, the prevalence of staphylococcal skin colonization in psoriatic patients was 36.6% (95% CI 18.3-54.8), vs. 5.1% in healthy subjects, (95% CI 3.0-8.5). In addition, MRSA appeared to colonize both the skin and nasal mucosa of psoriatic patients with a higher prevalence (pooled prevalence 8.6%) than healthy controls (2.6%), although this difference was not statistically significant (p=0.74)”. 

Original article, Abstract section:

“Overall we identified 21 eligible studies, of which 15 enrolled one or more comparison groups. The pooled prevalence of staphylococcal colonization in patients with psoriasis was 35·3% [95% confidence interval (CI) 25·0–45·6] on lesional skin and 39·2% (95% CI 33·7–44·8) in the nares. Patients with psoriasis were 4·5 times more likely to be colonized by S. aureus than healthy controls were on the skin (RR 5·54, 95% CI 3·21–9·57) and 60% more in the nares (RR 1·60, 95% CI 1·11–2·32). Cutaneous and nasal colonization by meticillin-resistant S. aureus also appeared higher in patients with psoriasis (pooled prevalence 8·6%) than in healthy controls (2·6%), yet the difference was not statistically significant (P = 0·74)”. 

Example 2:

Review: 

“Atefi et al. [29] also confirmed in a case-control study that psoriatic patients have a higher frequency of staphylococcal superantigens measured in serum. In particular, Toxic Shock Syndrome Toxin (TSST) was detected in 47% (20/41 cases) vs. 6% (1/28 controls), (p = 0.000), as well as Enterotoxins (A, B, D) were detected in 48.8% (21/41 cases) vs. 6% (1/21 controls), (p = 0.000)”

Original article, Abstract section: 

“TSST (toxic shock syndrome toxin) was detected in 47% (20/41) of cases and in 6% (1/28) of the controls with a significant difference. (P value = 0.000) Entrotoxins (A, B, D) were detected in the sera of 48.8% (21/41) of cases; and only 6 %(1/21) of controls, showed significant differences (P value = 0.000)”

Results that are non-significative should not be addressed in a review unless controversy is reported in that regard. The authors should extract the main findings in each manuscript cited.

RE: We improved all the aspects required, see section 1.1

I suggest the use of tables to summarize results, this will help to organize information and makes the manuscript text more fluent.

RE:Done, see table 1

The authors don’t mention the alterations in the epidermis that is also a risk factor for infection. A recent study shows that psoriatic plaques present epidermal barrier dysfunction and the use of phototherapy may improve the epidermal barrier in psoriatic patients (Montero-Vilchez T, et al., 2021. Epidermal barrier changes in patients with psoriasis: The role of phototherapy. Photodermatol Photoimmunol Photomed. 2021 Jul;37(4):285-292). Phototherapy is also one of the therapeutics in development for the treatment of Staphylococcus aureus skin infections, so may be one of the new therapeutic perspectives to be considered in the review.

RE: Done

Another gap in new therapeutic perspectives in this manuscript is nanotechnology, there is extended literature on the use of different nanoparticles for the treatment of Staphylococcus aureus infections, and a recent study present this technology also for drug administration in psoriatic patients (Petit RG, et al.; 2021. Psoriasis: From Pathogenesis to Pharmacological and Nano-Technological-Based Therapeutics. Int J Mol Sci. 2021 May 7;22(9):4983).

RE:Thanks for the advice, we added the suggested article. 

A new low-cost therapeutic perspective is also represented by phototherapy with a combination of two wavelengths, 460 nm and 405 nm, which showed to be effective in controlling MRSA infection in a mouse model. [70] Phototherapy also acts by restoring the barrier function of psoriatic lesional skin, with an improvement in both trans epidermal water loss (TEWL), stratum corneum hydration (SCH), pH, and elasticity. [71] This could contribute in our opinion to the reduction of the risk of colonization and infection by S. aureus, restoring physical skin barrier function and also indirectly acting on local microbiota alterations in the psoriatic plaque. [72]
Finally, the use of nanotechnology as a carrier for various drugs [73] could also help in controlling MRSA infections in psoriatic patients. Among the molecules that in our opinion could be more easily used in the near future, mainly due to their low cost, are Zinc oxide nanoparticles (ZnO-NPs), which act in vitro by producing reactive oxygen species and inhibiting amino acid synthesis in S. aureus strains [74], and nickel nanoparticles (NiNPs), which act by altering the structure of bacterial proteins, ba
cterial cell membranes and inhibiting DNA replication in MRSA. [75] However, the lack of human studies and the possible risk of increased penetration into the skin, with further impairment of the skin barrier, require further studies before use in psoriatic patients.

I find the topic of the review extremely interesting but the manuscript was not rigorously organized, and lack critical information concerning the pathology of psoriasis that could help understand the link between Staphylococcus aureus infections and psoriasis, whether psoriasis favors the colonization and infection or not; whether the bacteria enhance psoriasis or not, the mechanism that are or could be involved, etc. All the questions that do not find answers in the literature could be presented as interesting fields to explore in conclusion and perspectives.  

RE: We imporved the organization of the manuscript as required and added the statement in the conclusion and perspectives.

  1. Conclusions

To the best of our knowledge, there is a lack of crucial information in the literature regarding the correlation between psoriasis and S. aureus. Further studies are needed to better explore the role of staphylococcal infection and colonization in the pathogenesis of psoriasis, especially at the molecular level, potentially allowing new mechanisms to be discovered that can be used in the therapeutic approach.

Reviewer 2 Report

The article submitted for review has very great potential, and is a major contribution to systematizing current trends and research in the broadest sense skin infections. It is also relevant to the very rapidly developing research area about regenerative medicine.  The article cites several compounds used to treat skin infections including MRSA and their interrelationships and additional complications in patients with psoriasis. In my opinion, going with the theme of the article, I would have expected it to the article should be divided into the parts about different kind of the therapies used and described them in detail.

If the authors want to stay with the current form then several changes should be made: 

1. there should not be a section "results and discussions" - this a subsection is dedicated to research publications, not "perspective" which is only a literature review.In this paper authors should rather stay only with the subsections proposed as 2.3. , 2.4. etc.

2. Subsections 2.1. ; 2.2. did not present information about therapy against MRSA or skin infection for patient with psoriasis. They are rather a statistical summary obout the scale of the problems. 

This part succefully could be as introduction showing the enormity of the problem.  The reference to immunetherapy with only two drugs : ustekinumab or adalimumab as  a one sentence is a little bit too small when we writing about therapies and methods of the treatment. I would have expected a more in-depth analysis apo the use of the mentioned therapies rather than statistics apo infections and interrelationships. The other two subsections 2.3 and 2.4 are correct. 

3. There should not be a "materials and methods' chapter, this is reserved for article, research work not for literature review. 

Author Response

The article submitted for review has very great potential, and is a major contribution to systematizing current trends and research in the broadest sense skin infections. It is also relevant to the very rapidly developing research area about regenerative medicine.  The article cites several compounds used to treat skin infections including MRSA and their interrelationships and additional complications in patients with psoriasis. In my opinion, going with the theme of the article, I would have expected it to the article should be divided into the parts about different kind of the therapies used and described them in detail.

If the authors want to stay with the current form then several changes should be made: 

  1. there should not be a section "results and discussions" - this a subsection is dedicated to research publications, not "perspective" which is only a literature review.In this paper authors should rather stay only with the subsections proposed as 2.3. , 2.4. etc.

RE: Thanks for the useful indication, we adapted the manuscript as suggested as we want to maintain the type of perspectives

  1. Subsections 2.1. ; 2.2. did not present information about therapy against MRSA or skin infection for patient with psoriasis. They are rather a statistical summary obout the scale of the problems. 

This part succefully could be as introduction showing the enormity of the problem. 

The reference to immunetherapy with only two drugs : ustekinumab or adalimumab as  a one sentence is a little bit too small when we writing about therapies and methods of the treatment. I would have expected a more in-depth analysis apo the use of the mentioned therapies rather than statistics apo infections and interrelationships.

RE:We improved the suggested aspects:

Finally, other studies in the literature do not report an increase in serious staphylococcal infectious episodes in patients with moderate to severe psoriasis on therapy with ustekinumab [32] or adalimumab [33], suggesting that appropriate immunomodulatory therapy for psoriasis may reduce the risk of severe staphylococcal infections. These drugs, interleukin (IL) 12/23 and tumor necrosis factor (TNF) a inhibitors respectively, do not in fact act as immunosuppressants but rather selectively modulate the immune response involved in the pathogenesis of psoriasis. The resulting sustained reconstitution of the integrity of the skin barrier may reduce the risk of developing S. Aureus infections. However, these biological drugs may increase the risk of reactivation of latent chronic infections, as in the case of tuberculosis, HIV, HCV, and HBV. There are no other specific data in the literature on serious S. aureus infections in psoriatic patients undergoing therapy with biologics, however, their use is contraindicated in the case of serious infections.

The other two subsections 2.3 and 2.4 are correct. 

  1. There should not be a "materials and methods' chapter, this is reserved for article, research work not for literature review.

RE:We removed the materials and methods chapter 

Round 2

Reviewer 1 Report

The authors presented an improved manuscript, they followed the suggestions and answered all the questions.  Congratulations are in order. 

Please check for  "S. Aureus" and "S. aureus" in the manuscript the correct form is "S. aureus